# Symmetry breakdown of 4,4″-diamino-*p*-terphenyl on a Cu(111) surface by lattice mismatch

Qigang Zhong[1], Daniel Ebeling [2], Jalmar Tschakert[2], Yixuan Gao[3], Deliang Bao[3], Shixuan Du[3], Chen Li[4], Lifeng Chi[1] & André Schirmeisen[2]

Site-selective functionalization of only one of two identical chemical groups within one molecule is highly challenging, which hinders the production of complex organic macro-molecules. Here we demonstrate that adsorption of 4,4″-diamino-*p*-terphenyl on a metal surface leads to a dissymmetric binding affinity. With low temperature atomic force microscopy, using CO-tip functionalization, we reveal the asymmetric adsorption geometries of 4,4″-diamino-*p*-terphenyl on Cu(111), while on Au(111) the symmetry is retained. This symmetry breaking on Cu(111) is caused by a lattice mismatch and interactions with the sub-surface atomic layer. The dissymmetry results in a change of the binding affinity of one of the amine groups, leading to a non-stationary behavior under the influence of the scanning tip. Finally, we exploit this dissymmetric binding affinity for on-surface self-assembly with 4,4″-diamino-*p*-terphenyl for side-preferential attachment of 2-triphenylenecarbaldehyde. Our findings provide a new route towards surface-induced dissymmetric activation of a symmetric compound.

[1] Institute of Functional Nano & Soft Materials (FUNSOM), Jiangsu Key Laboratory for Carbon-Based Functional Materials and Devices, Soochow University, Suzhou 215123, P. R. China. [2] Institute of Applied Physics, Justus-Liebig University, Heinrich-Buff-Ring 16, 35392 Giessen, Germany. [3] Institute of Physics & University of Chinese Academy of Sciences, Chinese Academy of Sciences, Beijing 100190, P. R. China. [4] School of Environment and Civil Engineering, Dongguan University of Technology, Dongguan 523808, P. R. China. Correspondence and requests for materials should be addressed to D.E. (email: daniel.ebeling@ap.physik.uni-giessen.de) or to S.D. (email: sxdu@iphy.ac.cn) or to L.C. (email: chilf@suda.edu.cn)

Synthesis of complex macromolecules often remains a challenge, since it is extremely difficult to differentiate among the multitude of equal functional moieties that exist in an organic compound. For example, the selective substitution of only one C–H bond in a hydrocarbon derivative is a complex task, which relies on the choice of catalyst, steric hindrance, assistant directing units, or electronic activation. In a symmetric molecular structure with several similar or identical reactive groups, however, it will be almost a "mission impossible" to realize a completely site-selective functionalization. In those cases, purification becomes more crucial than the execution of the reaction to obtain the intentionally designed transformation. Thus, new methodologies beyond current organic synthesis are highly desirable. In recent years, the on-surface synthesis of molecular nanostructures has become a quickly developing field[1–16]. Numerous different coupling reactions have been applied to fabricate various molecular structures[15–20] that were not accessible via wet chemistry synthesis[21–23]. In such cases, on-surface reactions can implement hierarchical approaches[5,24–28], which involve the subsequent utilization of similar or different reaction mechanisms or take the advantage of the selectivity of certain types of reactions[29,30]. Consequently, on-surface synthesis benefits from extraordinary controllability over the molecular formation process. Until today, the conversion of same to different reactivity among identical functional groups of organic molecules has not been investigated.

Scanning probe techniques are suited for studying such local differences in reactivity since they provide the possibility of imaging single molecules. In particular, the functionalization of the tip of a low temperature atomic force microscope (AFM) with a single CO molecule enhances its lateral resolution and allows to identify the adsorption structures with atomic precision. This so-called "chemical bond imaging" technique was introduced in 2009 by Gross et al.[31], and has been used to image single molecules on surfaces, determine the adsorption orientation, follow reaction pathways, determine the absolute configuration of chiral compounds, etc[32–43].

Here, we report a process that enables the identification of subtle differences in the adsorption geometries of a symmetric compound on a surface, and exploit this feature to selectively modify the binding affinity of only one of two identical chemical groups. As the symmetric test molecule we use 4,4″-diamino-p-terphenyl (DATP), which has two identical amine end groups. In order to break the symmetry of the two end groups we use a reactive Cu(111) substrate, which has two features: it is incommensurate with the structure of the DATP molecule, while at the same time it shows a strong reaction affinity to amine groups. On Cu(111) we observe coexistence of two different types of adsorption structures for the DATP molecules, a symmetric and a dissymmetric one. The dissymmetric state is non-stationary, i.e., fluctuations are observed when the AFM tip is brought close to one end group of the molecule, resulting in a "fuzzy" image contrast. This serves as a fingerprint for the dissymmetric DATP. On a Au(111) substrate only a symmetric adsorption state is found. Comparison with dispersion corrected DFT-D2 computations reveals that a mismatch between the DATP molecules and the Cu(111) surface in combination with interactions with the subsurface atomic layer causes the asymmetric adsorption geometry. More interestingly, we are able to show that this has an influence on the local binding affinity of the adsorbed DATP. When adding a small amount of 2-triphenylenecarbaldehyde (TPCA) molecules on the surface, we observe preferential attachment to the fuzzy side of the dissymmetric DATP, leading to dissymmetric on-surface self-assembly. In contrast, symmetric attachment of TPCA to both end groups is found for the symmetrically adsorbed DATP. This demonstrates a viable route towards dissymmetric activation of a mirror-symmetric molecule

based on the balanced interplay of atomic structure and reactivity of a metal surface with aromatic molecules.

## Results and Discussion

**Adsorption structures of DATP on Cu(111).** In order to study the adsorption geometry of pristine DATP (see molecular structure in Fig. 1b) on Cu(111), the DATP molecules have been sublimed onto a cold substrate (below 100 K) using a home-build evaporation device[41,44,45]. All STM/AFM (STM: Scanning Tunneling Microscopy) scans have been performed at a low temperature of 5 K. Submolecular resolution is achieved by functionalizing the AFM tip with single CO-molecules[31]. An overview AFM frequency shift image illustrating adsorbed DATP on two adjacent terraces that are separated by a monoatomic step is depicted in Fig. 1a. The submolecular resolution AFM scans show that two distinct image contrasts for the adsorbed DATP molecules coexist on Cu(111). These two types are indicated by red (type I) and orange (type II) arrows. We find that 77.6% of the molecules are of adsorption type I, while 22.4% are of type II, which suggests that type I structures are energetically preferred.

High-resolution images of both structural types are depicted in Fig. 1c, d. The type I molecules are particularly interesting, since we observe fluctuations above one of the functional end groups, while the other end is stable during imaging. The fuzzy end of the dissymmetric molecules is visualized by the red arrows in Fig. 1. All these molecules align perfectly well with the crystallographic [11−2] direction (and their equivalent counterparts, i.e. [1−21] and [−211], see Fig. 1a). Interestingly, the fuzzy sides of the molecules on the surface (irrespective of the respective surface terrace) are always pointing towards a certain direction. In Fig. 1a, all molecules can be assigned with one of the three red arrows, while no molecule is pointing to the opposite direction (white dashed arrows). This reduction from six-fold to three-fold symmetry is caused by interactions with the subsurface atomic layer as discussed below[44,46]. Type II adsorption structures, on the contrary, appear to be symmetric.

The DATP molecules can switch fully reversibly between the two adsorption types I and II by manipulations with the AFM tip. It has been frequently observed that the molecules switch from type II to type I during constant height AFM scanning as shown in Fig. 1e, implying that type II should be energetically less favorable than type I. The scan was executed above a type II molecule (slow scanning direction is from bottom to top). After completing ~50% of the scan, a sudden switch of the adsorption structure from type II to type I occurred, which involves a rotation of the molecule by ~21°. We can also intentionally switch between the two adsorption structures by using the AFM tip as a manipulation tool, as demonstrated in the Supplementary Figs. 1, 2. Since we can deliberately switch between the two adsorption states by rotation, we conclude that the dissymmetric appearance of type I structures is not caused by a chemical modification of the molecule, i.e. DATP should be intact.

**Dynamics and energy barrier of the hopping mechanism.** Next we analyze the dynamics of the observed fluctuations above the type I adsorption geometry, causing the fuzzy image contrast. Figure 2a–d shows four constant height AFM scans of type I DATP molecules imaged at four different temperatures ranging from 5 K to 14 K. All images have been scanned with the same tip velocity, hence the fuzzy features in the four images directly reveal an increasing jumping rate with increasing temperature (see red dashed ovals). To quantify the jumping rates at different temperatures, the CO tip was placed over the fuzzy side of the molecule at a certain tip height with disabled STM feedback, and the tunneling current was recorded as a function of time

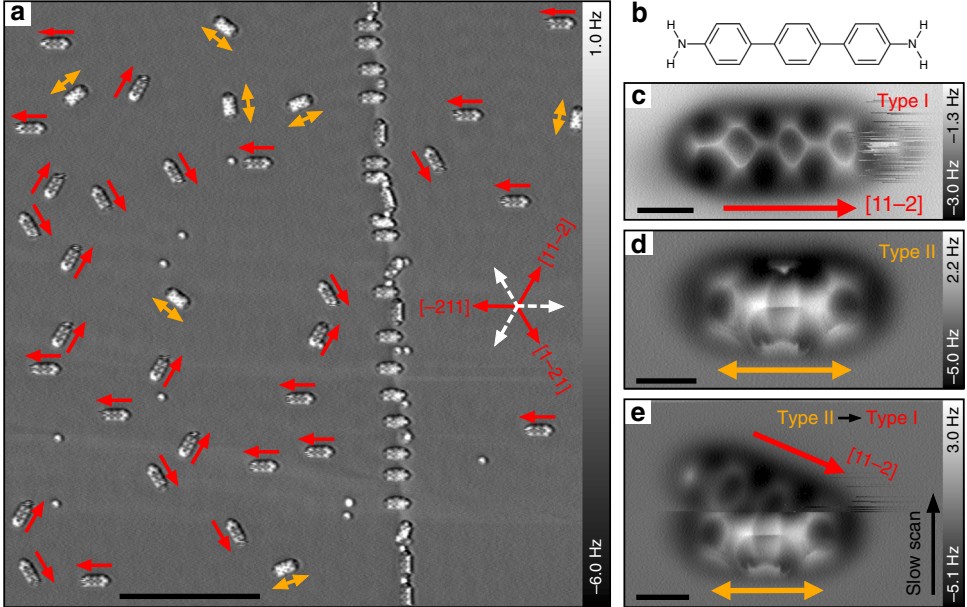

**Fig. 1** Different adsorption geometries of DATP on Cu(111). **a** AFM frequency shift image showing two terraces (upper terrace in right and lower in left part) separated by monoatomic step. The tunneling current was used for tip-sample feedback. Imaging parameters: gap voltage = 10 mV, tunneling set point = 100 pA. Type I and II adsorption structures are indicated by red and orange arrows. Red arrows are pointing towards the fuzzy side of the type I structures. Symmetry reduction from six-fold to three-fold is observed, i.e., orientations that correspond to the three dashed white arrows are not observed on neither terrace. **b** Chemical structure of DATP. **c**, **d** Constant height AFM zoom-in images of type I and type II adsorption structures. **e** Transition from type II to type I during constant height upward scan. Parameters: the tip-heights $\Delta z$ are **c** + 20 pm, **d** −85 pm, **e** −85 pm, relative to a STM set point of 100 mV, 10 pA on bare Cu surfaces. The +/− signs represent increase/decrease of the tip-sample distance. Scale bars: **a** 10 nm, **c**–**e** 0.5 nm

(see Fig. 2e–h). The graphs show a characteristic two-level tele-graph signal, which strongly depends on the temperature. The jump rates were determined by fitting a Poisson distribution to the calculated histograms of jump periods (see Supplementary Fig. 3)[47–49] and displayed in an Arrhenius type plot in Fig. 2i. The Arrhenius fit yields an attempt frequency of $e^{9.40\pm0.22}$ Hz and an energy barrier of 5.15 ± 0.13 meV. Furthermore, we analyzed the bias and the distance dependence of the observed fluctuations (c.f. Supplementary Figs. 4, 5). We found that the fluctuations are induced by the presence of the AFM tip, which presumably deforms the local energy landscape of the adsorption state.

**Mismatch between DATP and the Cu(111)/Au(111) surface**. High resolution scans of the topology of the type I DATP show that the observed fluctuations are related to lateral jumps of the molecule between two preferred adsorption sites that are sepa-rated by ~90 pm from each other. This is illustrated in Fig. 3a, where the AFM image of a type I DATP has been overlaid with two molecular models. Images of the Cu surface in the vicinity of the molecule with atomic resolution allows us to determine the absolute adsorption position of the molecules (Supplementary Fig. 6). The sketch in Fig. 3b summarizes our experimental findings. In the following, the two metastable adsorption sites are labeled as type IA (black) and type IB (red). The left amine group of type IA is located close to a Cu atom (top site). Due to the geometric mismatch between the structure of the molecule and the Cu lattice, the other amine group of type IA is located approximately above a fcc hollow site (i.e., hollow site without atom in layer beneath). For type IB, the right amine group is located above a top site, while its left group is located above an hcp hollow site (hollow site with atom beneath). As fcc and hcp hollow sites are known to have different reactivity, this leads to a difference in adsorption energy between the IA and IB types[44,46,50].

To characterize the subtypes IA and IB in more detail, we performed dispersion corrected DFT-D2 computations (see Fig. 3c, d). The absolute adsorption positions are in good agreement with our experimental findings. In both cases, a local energy minimum for the respective position on the Cu surface is found and, indeed, we find that one subtype is slightly preferred over the other (type IA is preferred by ~8 meV). Please note that the calculated energy difference between the two adsorption states is in remarkable agreement with the energy barrier determined from the experiments (see Fig. 2). Furthermore, the computations reveal that the adsorption structures are not planar, i.e., in the cases where the amine group is located above a Cu top site it is significantly bent down towards the Cu surface plane (by ~70 pm). This finding is underpinned by our 2D frequency shift vs distance measurements (see Supplementary Fig. 7 for details), which reveal vertical jumps of the amine group at the fuzzy side towards the AFM tip when the tip is in close vicinity to this group. Therefore, we conclude that the observed fuzziness at one side of the DATP molecules is caused by a switching/jumping between the two adsorption structures type IA and IB, triggered by the proximity of the CO-tip to the corresponding amine group.

These findings allow us to understand the observed on-surface orientation selectivity, i.e., the symmetry reduction from six-fold to three-fold of the type I structures (see Fig. 1). The different reactivities of hcp and fcc hollow sites on Cu(111) provoke a difference in adsorption energy between the subtypes IA and IB. Since the energy barrier between the two subtypes is low, the molecules will in general assume the energetically lower type IA state, where one of the two amine groups is significantly bent towards the Cu surface plane (left amine group in Fig. 3d). This bending is an indication of the onset of a bond formation between the amine group and the corresponding Cu surface atom, which, however, weakens the N–H bonds within the amine group. As a result, the binding affinity of this amine group will be increased

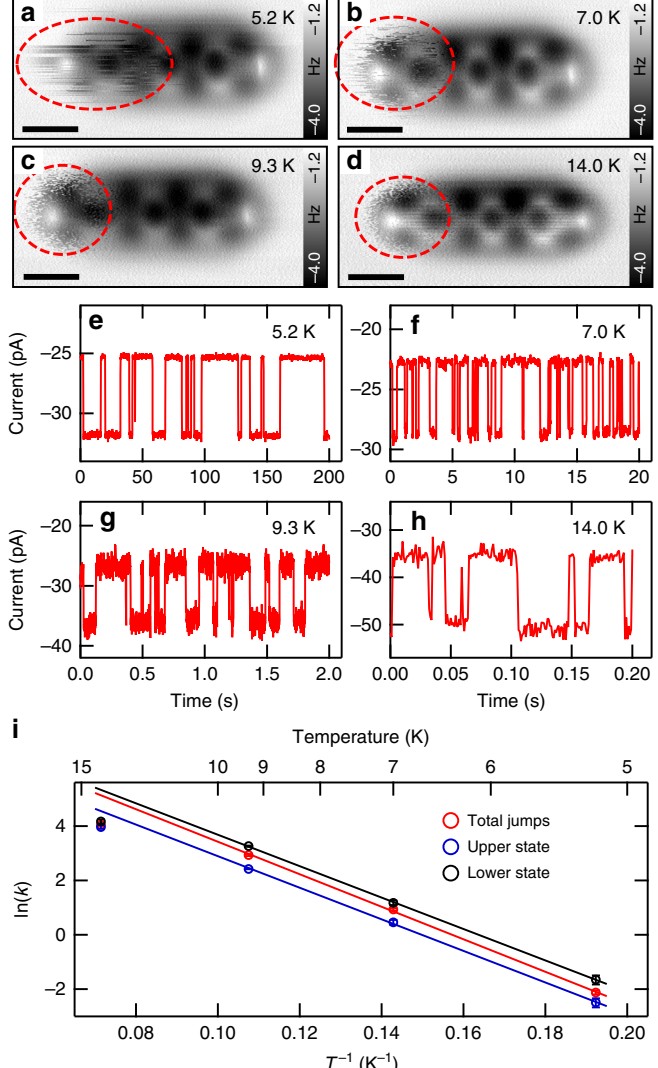

**Fig. 2** Dynamics and energy barrier of the observed hopping mechanism. **a–d** AFM images of type I DATP molecule taken at 5.2 K, 7.0 K, 9.3 K, and 14.0 K. The tip-heights $\Delta z$ are **a** −90 pm, **b** −85 pm, **c** −100 pm, **d** −90 pm, relative to a STM set point of 100 mV, 10 pA on bare Cu surfaces. **e–h** Current vs time traces recorded at the fuzzy end at 5.2 K, 7.0 K, 9.3 K, and 14.0 K. Parameters: $V_{bias} = 60$ mV, tip height $\Delta z = −10$ pm with respect to the STM tunneling set point of 100 mV, 10 pA on bare Cu(111) surface. **i** Arrhenius type plot of natural logarithm of the jumping rate $\ln(k)$ vs $1/T$. Displayed are the jumping rates for all jumping events (total jumps, red circles) and the rates for jumps into the upper (blue circles) and into the lower current states (black circles), respectively. An energy barrier ($E_\alpha$) of $5.15 \pm 0.13$ meV and a pre-exponential factor ($A$) of $e^{9.40\pm0.22}$ s$^{-1}$ for the total jumps is determined by fitting the Arrhenius' equation ( $\ln(k) = (−E_\alpha/k_B)(1/T) + \ln(A)$, where $k_B$ is the Boltzmann constant) to the right three points (at 5.2 K, 7.0 K, and 9.3 K). Jumps into lower state: $E_\alpha = 5.00 \pm 0.09$ meV, $A = e^{(8.69\pm0.16)}$ s$^{-1}$. Jumps into upper state: $E_\alpha = 4.98 \pm 0.04$ meV, $A = e^{(9.46\pm0.08)}$ s$^{-1}$. Please note, the data point at 14.0 K has not been taken into account since the observed jumping rate for this temperature exceeds the bandwidth of our tunneling amplifier setup. The vertical error bars in **i** are derived from the standard deviation of $k$ (see Supplementary Fig. 3). Scale bar: 0.5 nm

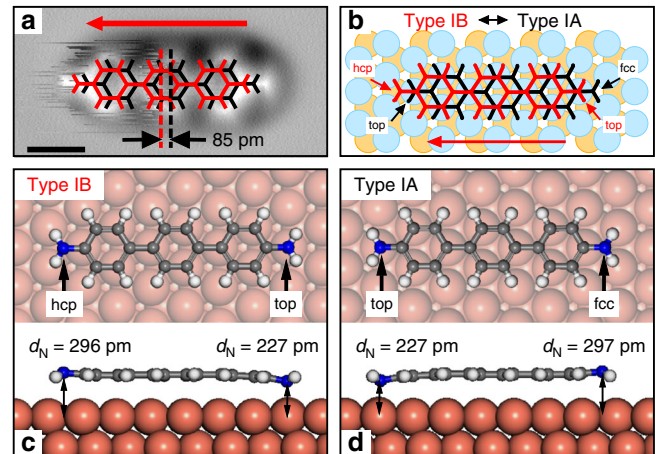

**Fig. 3** Adsorption structure of DATP type I on Cu(111). **a** AFM image of DATP type I fitted by two molecular models, namely type IA (black) and type IB (red). **b** Sketch of precise adsorption positions as determined by atomic resolution AFM scans (see Supplementary Fig. 6 for details). The two structures are displaced by 96 pm along the [11−2] direction. **c**, **d** DFT simulation of adsorption position type IA and type IB on Cu(111). Scale bar: 0.5 nm

by additional attractive forces between the CO-tip and the amino groups with higher binding affinity that trigger the jumping between type IA and type IB adsorption states.

The orientation of the fuzzy end group is predefined by the stacking order of the Cu(111) surface. Cu crystallizes in the face centered cubic lattice, thus, the close-packed atomic layers in (111)-direction follow an A-B-C stacking order, which dictates a certain helicity. Hence, the observed orientation of the fuzzy end is identical everywhere on the surface of the single crystal regardless of its specific terrace. Remarkably, in our case the selectivity of this mechanism is close to 100%, i.e., no exceptions have been observed (see red arrows vs dashed white arrows in Fig. 1a).

Further DFT calculations have been performed to gain more insight into the type II adsorption structure. In total, four different structures have been observed that are rotated by 14°, 17°, 21°, and 29° with regard to the [11−2] direction (see Supplementary Fig. 8). Overall, these structures are indeed energetically less favorable than type I structures (by 80–150 meV), which is in agreement with our experimental findings. The structure which bears the best resemblance with the experimental data is depicted in Fig. 4c. In this case, a symmetric adsorption geometry is observed with both amine groups bent down towards the surface plane (by ~70 pm). Following our line of arguments regarding the type I structures presented here, for type II molecules both end groups should behave similar with respect to their binding affinity.

In a control experiment we analyzed the adsorption geometry of DATP on Au(111), where we expect a significantly different behavior due to the different lattice constant of the substrate material. Corresponding DFT simulations and an AFM image of DATP on Au(111) are presented in Fig. 4b, d. We directly see that the mismatch is far less pronounced on this surface. The computations as well as our atomic resolution AFM scans (see Supplementary Fig. 9) reveal that both amine groups of the adsorbed DATP molecule are approximately located above Au atoms (top sites). Hence, the adsorption geometry is symmetric and, as expected, no fuzzy features are observed in the AFM scans. Furthermore, the computations reveal that the molecules are less bent on Au(111), which is presumably caused by its lower reactivity and explains why the molecules look less distorted in the AFM

with respect to its non-bent counterpart at the other side of the molecule. The CO molecule at the AFM tip is literally sensing this difference in binding affinity between the two amine groups. Thus, the observed fuzzy features in the AFM scans are generated

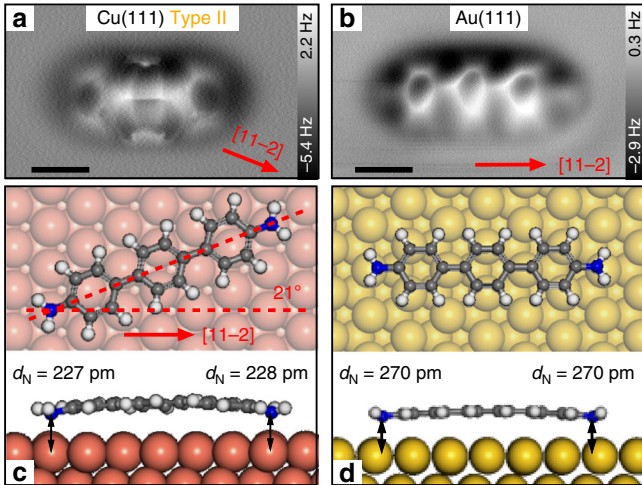

**Fig. 4** Adsorption structures of DATP type II on Cu(111) and Au(111). **a** AFM image of DATP type II on Cu(111). **b** AFM image of DATP on Au(111). **c, d** DFT simulations of corresponding adsorption structures on Cu(111) and Au(111), respectively. Scale bar: 0.5 nm

scans (cf. Figure 4a vs b). Hence, we are able to rationalize that the particular features (image distortions) observed above the middle carbon ring only on the Cu(111) surface (Fig. 4a) are caused by frustrated rotations of the corresponding ring.

**Enhanced binding affinity of fuzzy amino groups**. So far we demonstrated that a symmetric molecule can behave asymmetrically on a surface due to different adsorption positions. The main question remains, if such a weak effect can be exploited for inducing dissymmetric behaviors in, e.g., self-assembly or even in chemical reactions. Thus, a second molecular species TPCA was evaporated onto the cold substrate (<100 K). The ratio of DATP: TPCA was about 1.1 (Supplementary Fig. 10). An overview AFM image in Fig. 5a shows the formation of self-assembled molecular clusters. The zoom-in scan in Fig. 5b reveals that the molecules do not chemically react with each other at this temperature (<100 K). However, self-assembly of the two molecular species took place due to the formation of $O \cdots H - N$ hydrogen bonds (see Supplementary Fig. 11 for details). We observe individual type I and II of DATP molecules, as well as all the possible intermolecular combinations (Fig. 5a), including one DATP with one TPCA, and one DATP with two TPCAs.

In Fig. 5a we have marked the fuzzy end of DATP type I with red arrows. Evidently, the TPCA molecules exclusively connect to the fuzzy ends of the type I DATP molecules. In contrast, the TPCA molecules always connect to both sides of the symmetric type II DATP. Obviously, the bent-down end groups of the DATP (i.e., fuzzy end for type I, both ends for type II) show an enhanced binding affinity with the TPCA. This is corroborated by a comprehensive statistical analysis using STM overview scans analyzing a total of 1819 molecules (Supplementary Fig. 10). Among 89% combinations of single TPCA with single DATP type I molecule, the TPCA is located at the fuzzy end of DATP. For the symmetric type II structures, no such preference is detectable, with an almost equal ratio for connection to the two end groups (53%: 47%). Additional statistics are summarized in Table 1. Please note that, in total, an increased number of type II structures is counted after TPCA sublimation (57% after vs 22% before, see Fig. 1). Furthermore, there is a general preference for TPCA molecules to cluster with type II DATPs. Obviously, the presence of TPCA changes the adsorption energy landscape for the DATP molecules leading to a different ratio of molecular

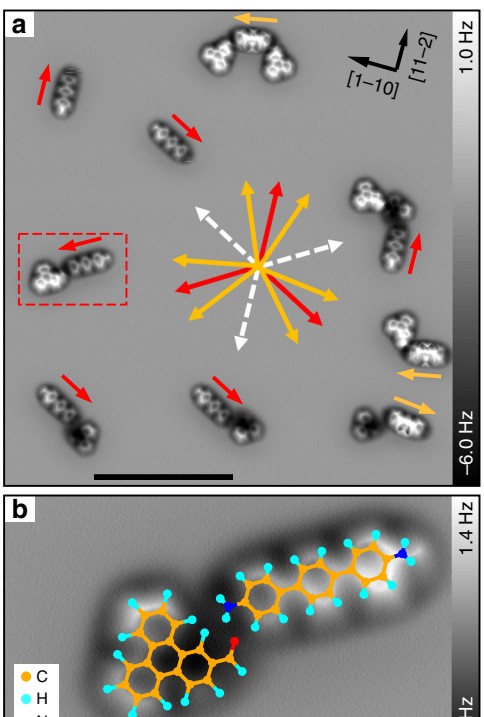

**Fig. 5** Self-assembly of DATP and TPCA molecules. **a** AFM overview image. Fuzzy sides of DATP type I molecules are indicated by red arrows. DATP type II structures are indicated by orange arrows. In total six orientations of DATP type II molecules exist, which are rotated by −21° and 21° with regard to the [11−2], [1−21], and [−211] directions (see also Supplementary Fig. 2). **b** AFM zoom-in image of a self-assembled DATP/TPCA cluster. Zoom area is indicated in **a** by red dashed box. The TPCA is non-covalently bonded to the fuzzy side of the DATP molecule (see Supplementary Fig. 11). Molecular models are added to the respective molecules. The yellow, light blue, blue and red dots represent C, H, N and O, respectively. Scale bars: **a** 5 nm, **b** 1 nm

geometry types. However, the dissymmetry effect for type I DATP remains untouched.

The preferred formation of certain DATP/TPCA clusters indicates that the amine groups which are bent towards the Cu surface have a higher binding affinity. The interaction between nitrogen lone pair and Cu seems to weaken the N–H bonds of the amine group or, at least, change the electron configuration at this side of the molecule. For the type I molecules this effect is even boosted by the non-stationary nature of this adsorption state. If a TPCA molecule diffuses closely towards a fuzzy end of a DATP molecule, the DATP can easily jump towards the TPCA molecule (due to the low energy barrier of ~5–6 meV) and form the connection via the hydrogen bond.

Essentially, we exploit the precise knowledge about surface adsorption geometry on metal surfaces to increase the binding affinity of only one end of a mirror-symmetric organic molecule. In our case, the observed bending of the DATP amine groups towards the Cu surface influences the binding affinity and thus induces this dissymmetry. Subsequent addition of TPCA molecules shows a clear influence on the on-surface self-assembly by dissymmetric clustering. We speculate that by carefully designing the molecules to increase the difference in binding energy or even create a chemical change at one end, one should be able to realize dissymmetric chemical reactions by using the concept we presented in this work.

**Table 1 Statistical analysis of self-assembled DATP/TPCA clusters**

| Total # of counted molecules DATP = 944 (TPCA = 875) | Single DATP | DATP occupied at one end | DATP occupied at both ends |
|---|---|---|---|
| DATP Type I = 406 (43%) | 254 (63%) | 99 (24%) Thereof: 88 at fuzzy side (89%) 11 at non-fuzzy side (11%) | 53 (13%) |
| DATP Type II = 538 (57%) | 115 (21%) | 385 (72%) Thereof: 205 at arrow tip (53%) 180 at other side (47%) | 38 (7%) |

## Methods

**Source of the molecules**. DATP (purity > 98%) and TPCA (purity > 98%) were purchased from TCI Company.

**AFM measurements**. The measurements were performed with a commercial combined low temperature AFM/STM (Scienta Omicron, Germany). All STM/AFM images were acquired at 5 K (Except for Fig. 2b–d) under ultra-high vacuum (base pressure < $1.0 \times 10^{-10}$ mbar). For STM imaging, the tip was connected to the ground while the sample was in contact with the bias voltage. For AFM imaging, a small offset gap voltage (a few uV) was used to minimize the tunneling current. The AFM imaging was realized with a force sensor based on the qPlus quartz tuning fork design[51]. Two different force sensors were used—for Fig. 4b, Supplementary Fig. 9a,b: resonance frequency $f_{res} \approx 19.4$ kHz, Q-factor ≈ 6300, oscillation amplitude Amp = 94 pm; for all other AFM images: resonance frequency $f_{res} \approx 27.0$ kHz, Q-factor > 10,000, oscillation amplitude Amp ≈ 60 pm (the oscillation amplitude for Fig. 1c is 143 pm). A PLL bandwidth of 10 Hz and a scanning speed of 380–630 pm/s were applied to all the AFM images. (Exceptions: scanning speeds are 1750 pm/s, 2000 pm/s, and 1130 pm/s for Supplementary Fig. 1, Fig. 5a, and Supplementary Fig. 9a, respectively.) To achieve submolecular resolution, the tip apex of the metal tip was functionalized with single CO molecules[52].

The Cu(111) surfaces were cleaned using multiple Ar+ sputtering and annealing cycles. To preserve the pristine DATP molecules, the DATP molecules were carefully evaporated onto cold Cu(111) surfaces ($T_{sample} < 100$ K). Further details about the home-built evaporation device can be found in refs. [41,44].

**Computational methods**. All DFT calculations are carried out using Vienna ab initio simulation package (VASP) with the projector augmented wave (PAW) method, and a generalized gradient approximation (GGA) in the form of Perdew–Burke–Ernzerhof (PBE) is adopted for the exchange–correlation functional. Long-range dispersion corrections have been taken into account within a DFT-D2 approach of Grimme. The energy cutoff of the plane-wave basis sets is 400 eV. A gamma-only k-point mesh is used to our calculation. A slab model is used with three Cu layers as the substrate. The vacuum layer is >15 Å. All atoms except the bottom Cu layer are fully relaxed until the net force is < 0.01 eV/Å.

**Data availability**. The authors declare that the data supporting the findings of this study are available within the paper and its supplementary information file.

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

## Acknowledgements

We acknowledge financial support from the Deutsche Forschungsgemeinschaft (DFG) via the GRK (Research Training Group) 2204 "Substitute Materials for Sustainable Energy Technologies". L.C. thank the National Science Foundation of China (21790053) and Ministry of Science and Technology of China (2017YFA0205002), S.D. thank the Chinese Academy of Sciences (XDPB0601) for the financial support. This project was also supported by the Laboratory of Materials Research (LaMa) of JLU and the LOEWE program of excellence of the Federal State of Hessen (project initiative STORE-E).

## Author contributions

D.E., L.C., and A.S. conceived the project and designed the studies. Q.Z., J.T., and D.E. performed the experiments and analyzed the data. Y.G., D.B., and S.D. performed the computations. C.L. contributed to the discussion of possible chemical reactions. The manuscript was written with contributions from all authors.

## Additional information

**Competing interests:** The authors declare no competing interests.

