## [Peer Review File · Nature Communications]

Editorial Note: This manuscript has been previously reviewed at another journal that is not operating a transparent peer review scheme. This document only contains reviewer comments and rebuttal letters for versions considered at Nature Communications. Mentions of prior referee reports have been redacted.

REVIEWERS' COMMENTS:

Reviewer #2:

The authors partially addressed the comments of the referees. However, the expression "activation" is not replaced in the entire manuscript by "increased binding affinity", e.g. see TOC Graphics or keywords. In addition, the focus of the introduction is still related to the on-surface synthesis. The presented results are of high quality and therefore I recommend publication in Nat. Comm. after considering the comments above.

Reviewer #3:

This manuscript ... [Redacted] has been revised carefully according to the comments. In my opinion, this manuscript will attract great interest both from physical chemists and organic chemists and is worthy to be published in Nature Communications at present form.

REVIEWERS' COMMENTS:

Reviewer #2:

The authors partially addressed the comments of the referees. However, the expression “activation” is not replaced in the entire manuscript by “increased binding affinity”, e.g. see TOC Graphics or keywords. In addition, the focus of the introduction is still related to the on-surface synthesis. The presented results are of high quality and therefore I recommend publication in Nat. Comm. after considering the comments above.

- ➔ The TOC graphic has been removed. We modified the keywords, abstract, and introduction parts accordingly. In two cases we kept the term “activation” but reworded the corresponding sentence to, e.g., “This demonstrates a viable route **towards** dissymmetric activation...” instead of “This demonstrates a viable route **for** dissymmetric activation...”. Everywhere else the expression “activation” has been replaced by “increased binding affinity”.

Reviewer #3:

This manuscript ... [Redacted] has been revised carefully according to the comments. In my opinion, this manuscript will attract great interest both from physical chemists and organic chemists and is worthy to be published in Nature Communications at present form.

- ➔ No changes required